# Evolution of Three-Dimensional Computed Tomography Imaging in Thoracic Surgery

**DOI:** 10.3390/cancers16112161

**Published:** 2024-06-06

**Authors:** Toyofumi Fengshi Chen-Yoshikawa

**Affiliations:** Department of Thoracic Surgery, Nagoya University Graduate School of Medicine, Nagoya 466-8550, Japan; tyoshikawa@med.nagoya-u.ac.jp; Tel.: +81-52-744-237-5

**Keywords:** robot-assisted thoracoscopic surgery, simulation, surgical guide, thoracic surgery, three-dimensional model, three-dimensional imaging, video-assisted thoracoscopic surgery

## Abstract

**Simple Summary:**

Lung cancer is the leading cause of death and the second most prevalent cancer worldwide. Recent improvements in radiological technologies enable the early detection of lung cancer. A minimally invasive approach has become a standard therapy for early-stage lung cancer, and therefore, the need for preoperative and intraoperative simulations has increased. Software evolution has broadened the possibilities of three-dimensional (3D) reformatting of computed tomography (CT), resulting in the wide use of 3D CT images in thoracic surgery. However, several issues need to be resolved for future improvement in current 3D CT imaging, such as the necessity of contrast-enhanced CT, the limitation of describing small branches of pulmonary vessels, and the static images not coping with surgical or deflated deformation. This narrative review summarizes the current situation and concerns of 3D CT imaging and its history and explores novel strategies to solve these issues.

**Abstract:**

Radiologic reconstruction technology allows the wide use of three-dimensional (3D) computed tomography (CT) images in thoracic surgery. A minimally invasive surgery has become one of the standard therapies in thoracic surgery, and therefore, the need for preoperative and intraoperative simulations has increased. Three-dimensional CT images have been extensively used, and various types of software have been developed to reconstruct 3D-CT images for surgical simulation worldwide. Several software types have been commercialized and widely used by not only radiologists and technicians, but also thoracic surgeons. Three-dimensional CT images are helpful surgical guides; however, in almost all cases, they provide only static images, different from the intraoperative views. Lungs are soft and variable organs that can easily change shape by intraoperative inflation/deflation and surgical procedures. To address this issue, we have developed a novel software called the Resection Process Map (RPM), which creates variable virtual 3D images. Herein, we introduce the RPM and its development by tracking the history of 3D CT imaging in thoracic surgery. The RPM could help develop a real-time and accurate surgical navigation system for thoracic surgery.

## 1. Introduction

Lung cancer is the leading cause of death and is the second most prevalent cancer worldwide. Recent improvements in radiological technologies enable the early detection of lung cancer. Minimally invasive approaches, such as video-assisted thoracoscopic surgery (VATS) and robot-assisted thoracoscopic surgery (RATS), have become standard therapies for early-stage lung cancer [1]. Pulmonary lobectomy and segmentectomy are the two main types of anatomical resections for lung cancer, and segmentectomy has recently been demonstrated to be a valuable alternative to lobectomy [2]. Anatomical segmentectomy is a more challenging procedure than lobectomy, since the anatomy of pulmonary segments is complex. New radiological technologies, such as thin-slice computed tomography (CT) and volume-rendering reconstruction for three-dimensional (3D) CT imaging, have allowed surgeons to plan procedures by analyzing radiological findings [3].

Three-dimensional CT imaging is a powerful tool for surgeons, and 3D CT images have recently been introduced into thoracic surgery [1,3,4]. In the past, 3D CT images were mainly used in the surgery of advanced tumors with resections of adjacent organs and/or large vessels as well as complex surgical procedures to better understand their anatomies. These images were created only by radiologists or technicians within a certain period. More recently, with the advancement of radiological imaging technology and the widespread use of thin-slice CT, small nodules can be frequently identified and resected for malignant tumors. Three-dimensional CT imaging has been used worldwide for enhanced precision surgery of these small nodules [3,4]. For example, it has been widely used in sublobar resections for preoperative markings, such as virtual assisted lung mapping (VAL-MAP) and indocyanine green (ICG) VAL-MAP [5,6,7]. In these methods, preoperative 3D CT images after marking is the key step for a precise surgical resection [8]. The JCOG0802/WJOG4607L Japanese large-scale randomized trial and the CALGB/Alliance 140503 trial conducted in the United States have shown the potential of sublobar resections for the curative treatment of early-stage non-small cell lung cancer [9,10]. Multiportal VATS, RATS, and uniportal VATS have become more common surgical techniques [1,2], where 3D CT images have been frequently used as surgical guidance for a minimally invasive resection of untouchable and invisible small tumors. Furthermore, various kinds of software have been developed worldwide and marketed mainly for preoperative and intraoperative simulations to reconstruct 3D CT images [11,12]. Several software types have been developed and widely used by radiologists, technicians, and thoracic surgeons in Japan [1,3,4].

Three-dimensional CT printing and more developed technologies, such as virtual reality [VR], augmented reality [AR], extended reality, and mixed reality, are using 3D CT images for surgical training, preoperative planning, and intraoperative assistance in thoracic surgery [8,13]. Thus, 3D CT imaging technology has widely entered the clinical practice of thoracic surgery. This narrative review summarized and discussed the current status and future development of this technology, focusing on the unresolved issues for future improvement and several potential solutions, such as the development of virtual dynamic 3D CT imaging.

## 2. Current Status of 3D CT Images in Thoracic Surgery

### 2.1. History of 3D CT Imaging

In the early days, surgical simulations were performed using 3D CT images reconstructed mainly by radiologists and technicians within a certain period based on selected cases. Three-dimensional CT images were created for a better understanding of complex anatomy in difficult surgeries with extended resections and safe surgical procedures of rare cases [1,14,15]. With the development of radiologic technologies, more precise 3D CT images can be created without difficulty even by thoracic surgeons, where 3D CT imaging has been widely and increasingly used mainly for preoperative surgical simulation and intraoperative surgical guidance in thoracic surgery [3].

Three-dimensional CT images are more frequently used for segmentectomy in minimally invasive thoracic surgery (Figure 1). They are useful for precisely understanding anatomy and free surgical margins, especially in segmentectomy. In 2010, Akiba et al. investigated the anatomical variations of pulmonary veins with 3D CT images, and they reported satisfactory results of the 3D CT images [16]. In 2011, Oizumi et al. also reported that preoperative 3D CT images showed a high accuracy rate of 98% in thoracoscopic segmentectomy [17]. Shimizu and Nagashima analyzed the segmental anatomical patterns of the right upper lobe using 3D CT in several cases [18,19]. After that, many thoracic surgeons confirmed the usefulness of this technique, mainly in minimally invasive surgery, such as multiportal VATS, uniportal VATS, and RATS [20,21,22,23]. Regarding the current surge of 3D imaging analyses, Nakazawa and his colleague reviewed the lung anatomy using 3D CT imaging [24]. The lung anatomy has been studied in detail by cadaveric and surgical confirmation, and several critical limitations were reported, such as postmortem modifications and difficulties procuring numerous cases. In contrast, 3D CT imaging, which is largely noninvasive and can be analyzed from a radiological standpoint, allows the reconstruction of lung anatomy and a more intuitive understanding of the spatial relationships between structures. Some studies on 3D CT imaging have enrolled substantial numbers of cases in a short period of time, including more than 1000 or 5000 cases [25,26].

### 2.2. 3D Imaging with a Wide Variety of Use

Three-dimensional CT imaging is used in various ways. It has been used for 3D CT volumetry for size matching in living-donor lung transplantation [1,3]. Three-dimensional CT angiography is useful for predicting branches of pulmonary vessels in donor lobectomies [1,3]. Three-dimensional CT imaging can be used for predicting metastatic lymph nodes and tumor invasiveness in primary lung cancer in retrospective clinical studies [27]. The relationship between lung volume and segmental anatomy has been deeply analyzed using 3D CT imaging [28]. It also allows for new analysis of the spatial relationship between structures, such as intersegmental planes and intersegmental veins [29]. Shibasaki et al. demonstrated that 3D CT volumetry could predict postoperative pulmonary function independent of the resected lobe [30]. The combination of 3D CT and ICG could perform minimally invasive thoracic surgery accurately and effortlessly, such as a precise resection of pulmonary sequestration [31,32]. In addition, Chang et al. performed VATS segmentectomy by obtaining 3D CT images with cone beam CT in a hybrid operating room [33,34]. They developed a dual image navigation method by clipping titanium clips on the surface of the intersegmental plane visualized by the intravenous injection of ICG and obtaining 3D CT with cone beam CT to confirm anatomically correct segmental boundaries [34].

Three-dimensional CT imaging is widely used for chest wall surgeries for various diseases, such as malignancy, trauma, and deformity [35,36,37]. Nakamura S. et al. reported that a precise surgical resection of lung cancer which invaded the chest wall was performed easily and accurately by intraoperative real-time navigation using 3D CT imaging [35]. The chest wall has a complex and dynamic structure with skeletal and soft tissue components, and therefore, 3D CT imaging is useful in chest wall surgeries, which generally consist of both resection and reconstruction [36,37].

Surgical training is one of the most useful ways to use 3D CT images. One of the first virtual reality (VR)–based simulators for training in VATS was developed by Solomon et al. [38]. This device was simple and concise, and it consisted of a standard laptop computer and a haptic feedback device to control the surgical equipment. Then, Jensen et al. developed the LapSim VR simulator for VATS lobectomy (Surgical Science, Gothenburg, Sweden). They also evaluated its effectiveness in simulating VATS lobectomy [39]. Haidari et al. also investigated the validity evidence for the modified VATS lobectomy modules for the LapSim simulator with haptic feedback (Surgical Science, Gothenburg, Sweden) [40]. Bedetti et al. then developed another simulator that created a VR curriculum, representing an evidence-based approach for VATS training programs (Simbionix Products, Surgical Science, Gothenburg, Sweden) [41]. Finally, Han et al. investigated the effectiveness of 3D displays in uniportal VATS training [42]. Study participants indicated that the 3D display was advantageous because it improved depth perception, and therefore, the operability of the endoscopic instrument. With the development of training devices, a variety of simulators have proven useful as training devices to assist a wide range of thoracic surgeons, including novices, to develop and maintain their skills in thoracoscopic surgery.

### 2.3. Currently Available 3D CT Software

A wide variety of 3D CT software has recently been developed worldwide, such as Synapse Vincent (Fujifilm Medical Co., Tokyo, Japan), Ziostation 2 and REVORAS (Ziosoft, Inc., Tokyo, Japan), Materialise Mimics Innovation Suite 23 (Materialise NV, Leuven, Belgium), Deepinsight platform (Neusoft Group Ltd., Shenyang, China), and IQQA-Lung (EDDA Technology, Princeton, NJ, USA) [11,12,28,43,44,45,46]. Hagiwara et al. reported that 97.8% of the 316 branches of the pulmonary artery, where 3D images were constructed using Synapse Vincent, were consistent with intraoperative findings [11]. In addition, Nia et al. performed 3D CT reconstructions using Synapse Vincent in 26 consecutive VATS pulmonary anatomical resections. They found that 3D CT images were useful in preoperative planning and intraoperative guiding to enhance the surgeon’s knowledge of the patient’s specific anatomy and to reveal anatomic variations [47]. Nakao et al. also reported that the 3D CT images reconstructed by REVORAS were in almost complete agreement with the intraoperative findings in 18 of 20 cases [12]. Cannone et al. presented their experience of 11 VATS anatomical resections performed after accurate preoperative planning with 3D reconstructions using a commercially available software, Materialise Mimics Innovation Suite 23 [43]. Moal et al. also reported 3D CT reconstruction for operative planning in nine cases of RATS segmentectomy using highly specialized software (Visible Patient, Strasbourg, France) [44]. The most important point for the general-purpose software is that thoracic surgeons, radiologists, and technicians can create 3D CT images by themselves. Around the same time, advances in radiological imaging and the widespread use of sublobar resection led to the broader use of 3D CT images to identify the location of small lung nodules and to obtain accurate margins for regional resections [5,6,7].

### 2.4. Three-Dimensional CT Imaging Recognized by Three Dimensions

A major advantage of 3D CT imaging lies in its ability to recognize anatomical structures better than two-dimensional (2D) imaging. The use of 3D–printed models has more advantages in confirming 3D CT images in three dimensions than 2D general monitors [48,49,50,51,52]. Although Liu et al. reported no significant difference between 3D–printed models and 3D CT images for experienced surgeons [48], most of the studies demonstrated that 3D–printed models improved various surgical outcomes more than 3D CT images on 2D monitors [49,50,51,52]. However, 3D–printed models are expensive and time-consuming, and it is difficult to generate 3D–printed models in all cases. It is useful in the first few cases to apply a new idea to actual clinical practice. For example, Chen et al. made 3D–printed models for surgical planning in the first case of a right-to-left inverted living-donor lobar lung transplantation, followed by multiple successful cases [53]. The development of a patient-specific, realistic, and reusable VATS simulator using 3D printing technology was also reported for a pediatric patient with an esophageal atresia and tracheoesophageal fistula [54]. A similar technique is the use of 3D stereoscopic vision using 3D polarized glasses or head-mounted displays. This AR imaging is another promising format for outputting 3D simulations, but only a few reports have been reported worldwide [55,56].

### 2.5. Current Issues to Be Resolved for Future Improvement

Three-dimensional CT has been one of the most innovative technologies in thoracic surgery in the last two decades [1,3], but several critical issues need to be resolved for future improvement. First, contrast-enhanced CT is generally required for the reconstruction of 3D images. It is practically difficult to reconstruct 3D images in patients with allergies to contrast agents, asthma, and low renal function. Second, small branches of pulmonary vessels and bronchi, which are generally within several millimeters in diameter, might not be detected with the current 3D CT technologies. Therefore, it is still mandatory for thoracic surgeons to check 2D CT images even after 3D CT images are reconstructed. Lastly, the lungs are soft organs, and what is seen intraoperatively differs from the 3D CT image reconstructed from the preoperative CT. For example, chest CT is performed preoperatively with the lungs being inflated, but the lung on the surgical side is deflated intraoperatively (Figure 2). In addition, the lung is retracted and deformed variably by the surgical instruments intraoperatively. The current simulator shows only a static inflated lung and does not follow the deformations caused by surgery or deflation. Thus, the ultimate 3D CT imaging should accommodate both deflated and surgical deformities. The resolution for this issue could eventually lead to successful surgical navigation in the future. Dealing with deformities as unmet needs is the next crucial step to realize precise surgical navigation.

## 3. Evolution of 3D CT Imaging

### 3.1. Three-Dimensional CT Images Reconstructed from Nonenhanced CT Data

Currently, 3D CT images are essential tools for precision surgery, but most software for reconstructing 3D CT images requires contrast-enhanced CT data. However, several studies have recently reported that 3D CT images can be created without contrast-enhanced CT data [57,58,59]. According to Nakazawa et al., they used Ziostation 2 (Ziosoft, Inc., Tokyo, Japan) to reconstruct 3D images from unenhanced CT data preoperatively in a case of multiple segmentectomies for a patient with a history of anaphylactic shock due to an allergy to iodinated contrast medium [57]. Intraoperatively, they rotated and resized the 3D images to match the actual intraoperative images and discussed any anatomical questions that arose within the team. They successfully accomplished a planned complex surgery, but they noted that the process of 3D image reconstruction was manual and required basic knowledge of lung anatomy. The disadvantage of this technique was that it was more time consuming than the usual semi-automated 3D reconstruction of the pulmonary vasculature based on contrast-enhanced CT images. Nakao et al. also performed a retrospective study of seven segmentectomies, in which 3D CT images were reconstructed using REVORAS based on unenhanced and contrast-enhanced CT data in each patient [58]. They stated that a 3D CT image simulation of a segmental resection from unenhanced CT has sufficient anatomical accuracy for practical use, but they also acknowledged that minor misidentifications were inevitable, which required caution. Chen et al. also arbitrarily selected 20 cases representing the most common segmental resections and conducted a retrospective pilot study to simulate 3D CT images [59]. In this study, they used a surgical planning assistance system developed using deep learning technology to validate a segmental resection with respect to its adjunctive role in surgical planning. They concluded that this artificial intelligence (AI) reconstruction algorithm was valuable in surgical planning for segmentectomy and that surgeons may achieve a high identification accuracy of anatomical patterns in a short time with AI reconstruction; however, the current accuracy levels of 3D CT images using this technology still leave room for improvement.

### 3.2. Development of Variable Virtual 3D Images

Three-dimensional CT imaging is a recently developed technology that is useful for thoracic surgery. However, current 3D CT technology only provides static images, so surgeons still had to rely on experience. For virtual dynamic images, there have been several reports of innovative uses of virtual reality simulations as training tools. However, their clinical applications are limited by their inability to incorporate individual patient-specific CT data into the generated images [60]. The Resection Process Map (RPM) was developed for thoracic surgery to address the unmet need to develop a new simulation system to simulate intraoperative deformations such as anatomical changes by a traction of lung parenchyma for better exposure of the surgical field (Figure 3) [61]. The RPM generates virtual dynamic images based on patient-specific CT data and was initially invented as a surgical simulation tool for liver surgery and later used as a useful surgical guide for anatomical lung resections [62].

The RPM is generated directly from patient-specific CT data and displays different resection paths along with high-quality visual maps as a surgical guide for anatomic lung resections. The system is designed to provide a semi-automatic system framework as well as to generate virtual images at high speed (Figure 3). Notably, the RPM can be operated effortlessly by thoracic surgeons. In the first report by Tokuno and Chen-Yoshikawa, the RPM accurately delineated 98.6% of the vascular branches and all bronchi, and the median time required for image acquisition was only 2 min (Figure 4) [61].

### 3.3. RPM Aiming at Real-Time Surgical Guide

In the second report by Tokuno and Chen-Yoshikawa, the versatility of the RPM was confirmed by testing its ability to synthesize virtual images with actual surgical video in representative anatomical surgical resection procedures (five lobectomies and four segmentectomies) [63]. In this report, the RPM successfully generated semi-automated virtual dynamic images from patient-specific CT data. Furthermore, when virtual images were superimposed on selected video clips from surgical videos, no significant differences were observed. This study was conducted in a retrospective setting. However, it demonstrated the potential of the RPM to be used clinically as a preoperative and intraoperative simulation, leading to the possibility of real-time navigation for clinical use in thoracic surgery. Another major application of 3D CT imaging is surgical education and training, and therefore, one of the strengths of this study was the educational aspect of explaining important surgical anatomy and crucial surgical techniques through superimposed images [8]. With the accumulation of surgical cases with more diverse techniques of anatomical surgical resection, a new surgical atlas with surgical videos and corresponding virtual dynamic 3D images will be created.

The first experience with the RPM in a real clinical setting was reported by Kadomatsu and Chen-Yoshikawa based on the results of a retrospective study [64]. In this report, the RPM was used in 13 anatomical resections by RATS, suggesting that intraoperative use of the RPM may prevent misidentification of lung structures and contribute to safe surgery. One of the advantages of the RPM is that it allows the surgeon to confirm the internal structures. Another advantage is that the structures that remain covered by lung parenchyma can be confirmed in advance, thereby reducing the risk of damaging critical structures. In addition, there are three points that should be considered for further improvement. First, manipulation of the RPM requires a manipulator who is familiar with the anatomy of the lungs. If the RPM can be operated with a non-contact sign, a console surgeon or assistant may be able to operate it, eliminating the need for a dedicated RPM operator. Second, the accuracy of vascular and bronchial depiction depends on the quality of preoperative CT. Pulmonary vessels with small diameters tend not to be delineated when constructed in 3D and need to be recognized by thoracic surgeons [12,24,64]. Third, because each lobe is segmented and reconstructed in the process of generating RPM images, all lobes are completely separated on the RPM. Thus, the RPM images do not reflect the actual depiction between the lobes.

### 3.4. RPM with Deformation by Deflation

The lung is a soft organ, and what is seen intraoperatively differs from the preoperative CT. The lung deforms dramatically during an operation by mobilization and ventilation. The current 3D CT imaging shows a static inflated lung when a chest CT scan is taken before surgery, and it does not follow the deformations caused by surgery or deflation. Therefore, it is essential for variable virtual 3D images to accommodate deflated and/or surgical deformities. Surgical deformation can be dealt using an RPM, but deformation by deflation should be coped with using another novel algorithm.

After surface deformation analysis of collapsed lungs using model-based shape matching in several experimental studies by Nakao et al., a novel algorithm for deflated deformation was developed [65,66]. First, CT images of the lungs of 11 live beagle dogs were acquired at different bronchial pressures to analyze the deformation of the collapsed lungs [65]. Then, CT images in patients with lung cancer were obtained by cone-beam CT in the inflated and deflated states to register and analyze the shapes of the lungs [66]. Using this lung deflation simulation algorithm, 3D CT images of the deflated lungs can be predicted only on the basis of preoperative CT images obtained during the inflated phase of respiration. A preliminary study was performed to retrospectively compare the intersegmental line predicted by the newly developed lung deflation simulation algorithm with that delineated by the intravenous administration of ICG [67]. Among 16 patients who underwent pulmonary segmentectomy, the concordance rate of these intersegmental lines was complete in twelve patients, partial in three patients, and discordant in one patient. The agreement rate for pulmonary intersegmental lines was 75%. In summary, the lung deflation simulation algorithm provided a new surgical guide in addition to those currently in use. A prospective study using this new deflation simulation algorithm is currently underway at the author’s institution.

Real-time surgical guidance is one of the goals for surgical navigation and has recently been tried in various fields of surgery. In 2023, first-in-human real-time AI-assisted instrument deocclusion during AR robotic surgery was reported [68]. Furthermore, the same group conducted first-in-human AI-assisted AR robotic lung surgery in two lower lobectomies with RATS [69]. Both reports were still preliminary reports, but these kinds of challenges would guide the future of navigation surgery in the field of thoracic surgery.

## 4. Future Perspectives

In the last two decades, 3D CT images have been reconstructed without difficulty and have been widely used in thoracic surgery. Moreover, several studies have reported that 3D CT images can be generated without contrast-enhanced CT data. However, the current 3D CT images are static and do not correspond to a surgical and deflated deformation. An algorithm to incorporate surgical deformations and deflated deformations of the lungs was developed to resolve this unmet need in the clinical practice of thoracic surgery. This algorithm has started to be clinically evaluated retrospectively and prospectively. It is highly expected that the current static 3D CT images can be turned into dynamic virtual 3D images for surgical navigation and more detailed surgical simulations in the near future. In addition, the next challenge is to automate all operations for rapid use in accordance with surgical flow. The first step would be selecting crucial surgery scenes and matching the images from there. A system that can handle deformations specific to surgery, such as pulling and inverting, should be developed. Validation of the more detailed algorithms developed this way would also be an essential step. Lastly, hurdles, such as the high cost and the time-consuming nature of surgery, should be solved with a novel technical breakthrough by surgeons and researchers.

## 5. Conclusions

Three-dimensional images have been developed and widely used as a useful surgical guide in thoracic surgery. However, current 3D images still have issues that need to be resolved for future improvement. One of them is that they provided only still images, which are different from the intraoperative views. Lungs are soft and variable organs that can easily change shape due to intraoperative inflation/deflation and surgical procedures. A variable virtual 3D imaging technique has been developed to encounter this unmet need. This novel technique was introduced by tracking its history and development in this narrative review. Furthermore, the history and current status of 3D CT images were summarized. The findings of this review could help surgeons and researchers update their understanding of 3D imaging in thoracic surgery and develop new strategies for improving thoracic surgery.

## Figures and Tables

**Figure 1 cancers-16-02161-f001:**
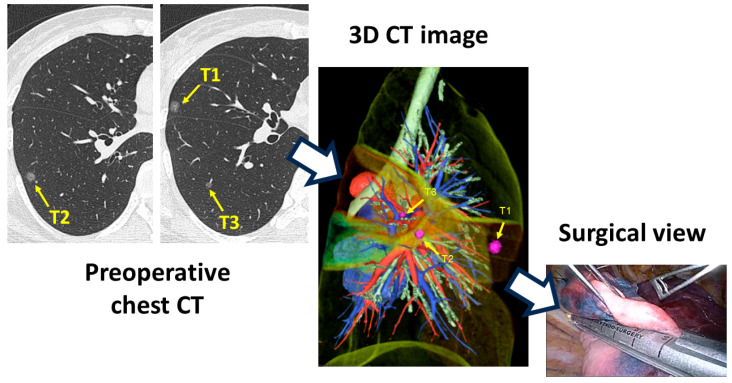
The common use of three-dimensional (3D) computed tomography (CT) image.

**Figure 2 cancers-16-02161-f002:**
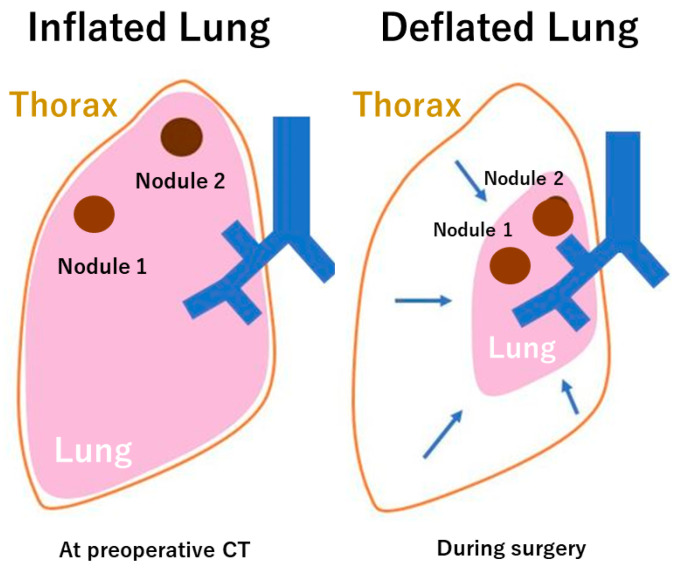
The schema of inflated lung at preoperative computed tomography (CT) and deflated lung during surgery.

**Figure 3 cancers-16-02161-f003:**
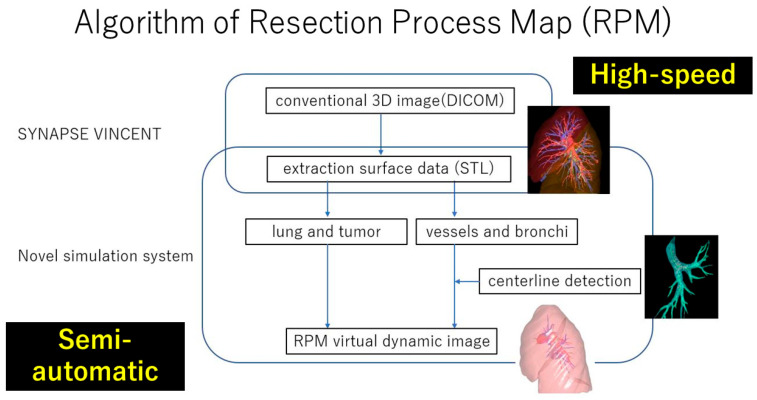
Algorithm of Resection Process Map (RPM).

**Figure 4 cancers-16-02161-f004:**
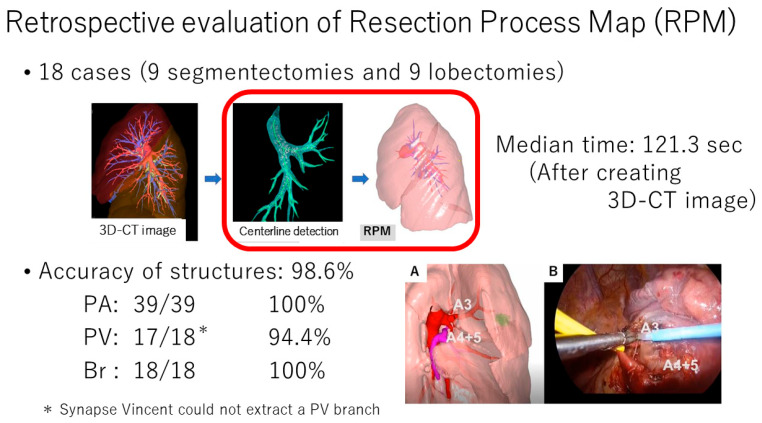
Retrospective evaluation of Resection Process Map (RPM). The RPM image (**A**) and corresponding intraoperative view (**B**).

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
