# Peer review of "Evolution of Three-Dimensional Computed Tomography Imaging in Thoracic Surgery"

_cancers, 2024, doi:10.3390/cancers16112161_

Round 1
Reviewer 1 Report
Comments and Suggestions for Authors
This is a very educational review that comprehensively covers the history of 3D-CT imaging in thoracic surgery, current preoperative simulation, and its use in education. In the latter half of the review, the status of development on remaining issues (3.1-3.4) and future perspectives (4) are presented, which is very meaningful. In particular, the authors' group's energetic research work in this area is a great learning experience for readers. It would be more attractive for readers if the following reports of real-time AI-enabled augmented reality at the real-time surgical guide from the group in the Netherlands/Belgium were also introduced herein.
https://www.ncbi.nlm.nih.gov/pmc/articles/PMC11022222/
https://www.jtcvstechniques.org/article/S2666-2507(24)00170-6/fulltext
minor point: The number '73' is missing in L494, P11
Author Response
Reply to the reviewer 1,
(Comment)
This is a very educational review that comprehensively covers the history of 3D-CT imaging in thoracic surgery, current preoperative simulation, and its use in education. In the latter half of the review, the status of development on remaining issues (3.1-3.4) and future perspectives (4) are presented, which is very meaningful. In particular, the authors' group's energetic research work in this area is a great learning experience for readers. It would be more attractive for readers if the following reports of real-time AI-enabled augmented reality at the real-time surgical guide from the group in the Netherlands/Belgium were also introduced herein.
https://www.ncbi.nlm.nih.gov/pmc/articles/PMC11022222/
https://www.jtcvstechniques.org/article/S2666-2507(24)00170-6/fulltext
minor point: The number '73' is missing in L494, P11
(Answer)
Thank you so much for the instructive suggestions. We made corrections according to the reviewer’s suggestions. Since one of the suggested references still seems “just accepted” or “in press”, we cannot find the paper in PubMed. However, on the basis of the reviewer’s kind comments, we added some comments in the 3.4 paragraph in the manuscript as follows:
“Real-time surgical guide is one of the goals for surgical navigation and has recently been tried in various fields of surgery. In 2023, first-in-human real-time AI-assisted instrument deocclusion during AR robotic surgery was reported [68]. Furthermore, the same group conducted first-in-human AI-assisted AR robotic lung surgery in two lower lobectomies with RATS [69]. Both reports were still preliminary reports, but these kinds of challenges would navigate the future of the navigation surgery in the field of thoracic surgery.”
[68] Hofman J, Backer PD, Manghi I, Simoens J, Groote RD, Bossche HVD, D’Hondt M, Oosterlinck T, Lippens J, Praet CV. First-in-human real-time AI-assisted instrument deocclusion during augmented reality robotic surgery. Healthc Technol Lett 2023; 11: 33-39.
[69] Sadeghi AH, Mank Q, Tuzcu AS, Hofman J, Siregar S, Maat A, Mottrie A, Kluin J, Backer PD. Artificial intelligence-assisted augmented reality robotic lung surgery; navigating the future of thoracic surgery. JTCVS Tech 2024 in press.
In addition, we also revised the manuscript according to the comments by the Editorial office.

Reviewer 2 Report
Comments and Suggestions for Authors
The manuscript was well written with good language and layout. However, the entire article only subjected on lung surgeries, which is not the only implementation of 3-D imaging device in thoracic surgery. As we know, 3-D imaging reconstruction has already been used in chest wall surgeries for trauma and deformities for some while with satisfactory results. I suggest you can add some paragraphs about these.
Author Response
Reply to the reviewer 2,
(Comment)
The manuscript was well written with good language and layout. However, the entire article only subjected on lung surgeries, which is not the only implementation of 3-D imaging device in thoracic surgery. As we know, 3-D imaging reconstruction has already been used in chest wall surgeries for trauma and deformities for some while with satisfactory results. I suggest you can add some paragraphs about these.
(Answer)
Thank you so much for the kind and meaningful suggestions. As we stated in the 2.2 section, 3D-CT imaging has been used in various ways in thoracic surgery, such as lung surgery and lung transplantation. As the reviewer suggested, this technique is also widely used in chest wall surgery. Therefore, we added some more descriptions in the manuscript as follows:
“3D-CT imaging is widely used for chest wall surgeries for various diseases, such as malignancy, trauma and deformity [35-37]. Nakamura S, et al. reported that a precise surgical resection of lung cancer which invaded chest wall was performed easily and accurately by intraoperative real-time navigation using 3D-CT imaging [35]. The chest wall has a complex and dynamic structure with skeletal and soft tissue components, and therefore, 3D-CT imaging is useful in chest wall surgeries, which generally consists of both resection and reconstruction [36, 37].”
35: Nakamura S, Hayashi Y, Kawaguchi K, Fukui T, Hakiri S, Ozeki N, Mori S, Goto M, Mori K, Yokoi K. Clinical application of a surgical navigation system based on virtual thoracoscopy for lung cancer patients: real time visualization of area of lung cancer before induction therapy and optimal resection line for obtaining a safe surgical margin during surgery. J Thorac Dis 2020; 12: 672-679.
36: Young JS, McAllister M, Marshall MB. Three-dimensional technologies in chest wall resection and reconstruction. J Surg Oncol 2023; 127: 336-342.
37: Pontiki AA, Natarajan S, Parker FNH, Mukhammadaminov A, Dibblin C, Housden R, Benedetti G, Rhode K, Bille A. Chest wall reconstruction using 3-Dimensiona; printing: functional and mechanical results. Ann Thorac Surg 2022; 114: 979-988.
In addition, we also revised the manuscript according to the comments by the Editorial office.

Round 2
Reviewer 2 Report
Comments and Suggestions for Authors
You have already revised accordingly.